# EMGBench: Benchmarking Out-of-Distribution Generalization and Adaptation for Electromyography

**Jehan Yang**[*]
Carnegie Mellon University
Pittsburgh, PA 15213
jehan@cmu.edu

**Maxwell Soh**[*]
Carnegie Mellon University
Pittsburgh, PA 15213
msoh@andrew.cmu.edu

**Vivianna Lieu**
Carnegie Mellon University
Pittsburgh, PA 15213
vlieu@andrew.cmu.edu

**Douglas J Weber**
Carnegie Mellon University
Pittsburgh, PA 15213
dweber2@andrew.cmu.edu

**Zackory Erickson**
Carnegie Mellon University
Pittsburgh, PA 15213
zerickso@andrew.cmu.edu

## Abstract

This paper introduces the first generalization and adaptation benchmark using machine learning for evaluating out-of-distribution performance of electromyography (EMG) classification algorithms. The ability of an EMG classifier to handle inputs drawn from a different distribution than the training distribution is critical for real-world deployment as a control interface. By predicting the user's intended gesture using EMG signals, we can create a wearable solution to control assistive technologies, such as computers, prosthetics, and mobile manipulator robots. This new out-of-distribution benchmark consists of two major tasks that have utility for building robust and adaptable control interfaces: 1) intersubject classification, and 2) adaptation using train-test splits for time-series. This benchmark spans nine datasets, the largest collection of EMG datasets in a benchmark. Among these, a new dataset is introduced, featuring a novel, easy-to-wear high-density EMG wearable for data collection. The lack of open-source benchmarks has made comparing accuracy results between papers challenging for the EMG research community. This new benchmark provides researchers with a valuable resource for analyzing practical measures of out-of-distribution performance for EMG datasets. Our code and data from our new dataset can be found at `emgbench.github.io`.

## 1 Introduction

Electromyography (EMG) sensors detect muscle and motor neuron activity from the body, allowing for wearable gesture-based control of robots or devices. Particularly, EMG sensors can be used to sense intended hand or other body movements from people who are unable to move parts of their body due to injury or neurodegenerative disease [1–5]. For people with upper or lower limb amputations, EMG-based prosthetic arms or legs can be controlled using the residual muscles from the remaining limb after amputation [2]. Additionally, for people with paralysis from stroke or spinal cord injury (SCI), EMG sensors can detect motor intent based on residual muscle fiber activity [4, 3, 5].

Several EMG datasets have been made publicly available, although many of them differ in regards to the hardware used and the placements of the sensors [2, 6–10]. Due to these common differences between EMG control interfaces, it is important to evaluate multiple EMG datasets to assess the classification accuracy and hardware-agnostic nature of machine learning-based techniques [11, 12].

---

[*]These authors contributed equally to this work.

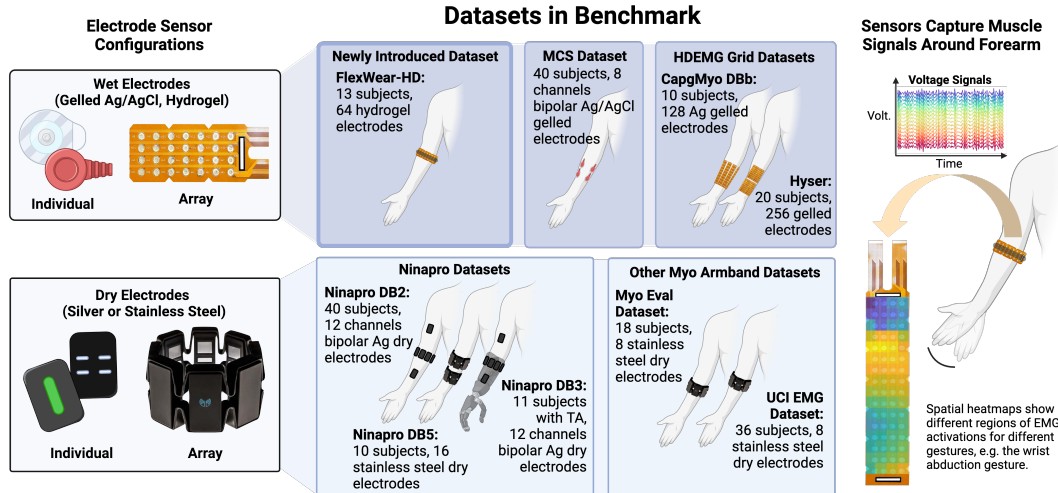

Figure 1: **Electrode configurations for various datasets and generalization task.** (Left) We show the main categories of electrode configurations used: dry electrodes and wet electrodes. These categories can be further separated into individually placed electrodes and electrode arrays. (Middle) The electrode configurations and placements used are shown for each dataset. We include nine total EMG datasets. Ninapro DB3 includes subjects with transradial amputations (TA). (Right) We show that voltage signals from the arm are detected using EMG sensors and illustrate that gesture-specific patterns of muscle activity can occur.

However, there is not yet a standardized benchmark for evaluating machine learning algorithm-based classification for EMG. This gap significantly impacts the standardization of results achieved from learning-based classification of EMG datasets, making results from machine learning papers that test on EMG difficult to compare. Furthermore, the absence of benchmarks designed to assess practical generalization and adaptation tasks, particularly those involving inter-subject performance evaluation, represents a notable gap in the current research landscape. In this work, we define and benchmark *generalization* as the ability of a model to classify the gestures of a subject without using any of their data for training, and *adaptation* as the ability of a model to personalize by fine-tuning using initial data from a subject after being pretrained on data from other subjects.

Improving performance on out-of-distribution subject generalization and adaptation-based tasks could significantly streamline the setup process for EMG interfaces, making them more accessible and easier to use for new users. For example, by benchmarking generalization tasks such as intersubject classification [13, 14], we can evaluate the performance of algorithms trained on large datasets to generalize to new subjects, enabling new users to control an interface without requiring additional data collection. In addition, by evaluating adaptation by fine-tuning using an initial subset of data from a new user [15, 14], we can determine the minimum amount of labeled data needed from a new user to personalize a model and achieve high gesture recognition accuracy [16]. For the aim of creating a benchmark for out-of-distribution subject generalization and adaptation for EMG, while including some of the most popular EMG datasets, we have also curated a number of datasets that have been demonstrated to have strong performance for out-of-distribution generalization and adaptation [17, 18, 7]. In all, we present the following contributions:

**The first open-source EMG benchmark** This benchmark presents a codebase for standardized evaluation of machine learning models on 9 curated EMG datasets for out-of-domain generalization and adaptation tasks. The codebase is available at `https://github.com/jehanyang/emgbench`.

**New dataset with wearable EMG sensor** We present data using an easy-to-wear, reusable, high-density EMG sensor. Results evaluating the generalizability between subjects in this dataset show high classification accuracy.

**Benchmarking results across tasks** Benchmarking results for a range of machine learning models and data preprocessing techniques across multiple generalization and adaptation tasks.

## 2 Background and Related Work

### 2.1 EMG signals

Although non-invasive EMG sensors are placed on the skin, these sensors can readily record significant voltage signals caused by the changes in voltage that occur following motor neuron action potentials. Muscles can amplify the biological voltage changes initiated by motor neurons [19]. This is because of the high number of moving ions during muscle fiber action potentials compared to neuronal action potentials, and the high number of muscle fibers activated per motor neuron, ranging from around 100 to as many as 1000 [19]. These changes in voltage signals can be detected by skin electrodes, with both dry and wet electrodes used in common non-invasive EMG devices, as shown in Figure 1. More details about EMG signals are in Appendix A.1.

### 2.2 EMG as a control interface

EMG sensors have demonstrated the capability to detect signals from the muscles in an amputated forearm [20], enabling high-dimensional control of prosthetic arms by leveraging residual muscle activity [21]. Furthermore, as an alternative to motion-based interfaces [22, 23], in individuals with clinically motor complete cervical spinal cord injuries resulting in hand paralysis, machine-learning algorithms have shown promise in predicting voluntary hand gesture intentions at the individual finger level, given EMG signals from seven subjects with paralysis [4].

EMG often faces non-stationary signals that historically have made generalization difficult for learning-based methods. However, these instances present a unique opportunity for domain adaptation methods. EMG signals experience several phenomena that cause **concept shifts**, altering the conditional probability of labels given the inputs from the training set to the test set [24]. The mechanisms causing this phenomenon include variations in muscle locations [25], arm sizes [26], skin impedances [27], and electrode placements [28]. By accounting for some of these mechanisms through fine-tuning or other methods, adaptation can potentially maintain or improve classification performance between subjects.

### 2.3 Classification over EMG datasets

Extensive research has focused on training machine learning models for EMG-based gesture classification, utilizing both publicly available datasets [18, 29–31] and novel datasets collected by the researchers [7, 1, 32, 17, 33–36, 14]. While many studies report results based on randomized train-test split accuracy [31, 36, 37] and k-fold cross-validation (KFCV) [38, 32, 39–41], where data from the training, validation, and test sets may be randomly sampled from the same subjects, such approaches may not accurately reflect real-world scenarios. In practice, it is often desirable for the validation and test sets to comprise data collected either temporally after the training data from the same subject (termed train-test splits for time series, or TSTS), or from a subject entirely excluded from the training set, as evaluated using leave-one-subject-out cross-validation (LOSO-CV). These data splitting strategies provide more robust assessments of model performance by introducing out-of-distribution generalization challenges. We present a categorization on how several other EMG classification papers split their data in Appendix Table 10.

In the case of TSTS, we test with data collected after the training set, which introduces potential distribution shifts due to factors such as variations in gesture execution, fatigue [42, 43], perspiration [44], electrode displacement on the skin [45], drying or changes in ionic concentrations of hydrogel or electrolyte gels [46], and variations in electrode adherence on the skin [47]. Similarly, LOSO-CV introduces variability stemming from inter-individual differences in body size, muscle morphology [25], and differences in skin impedance and adipose tissue distribution [27]. Studies employing randomized or mixed data splits, where evaluation data may precede training data, risk reporting artificially inflated accuracies that fail to reflect true generalization capabilities in practical EMG classifier deployments. To facilitate benchmarking, a standardized approach to EMG sample window extraction from raw data can ensure consistent data preprocessing across studies [48, 11]. This methodology enables more reliable comparisons of classification performance. A detailed analysis of prior work on EMG generalization and adaptation is provided in Appendix A.5.

# 3 Datasets

Several datasets have been released for classification of gestures using EMG. Between them, there are a large variety of experimental conditions and data collection protocols: some of these variations include the amount of time and repetitions used in cues for the participant to perform gestures [2, 49, 32, 30], different numbers of participants, and different devices used to collect data [2, 49, 8]. Figure 1 illustrates the sensor configurations and the number of subjects for each dataset, while Table 9 summarizes the amount of time the participant is cued to perform gestures. Further detail about each dataset is included in Appendix A.7 as well as in Table 6.

**Ninapro**   One of the most popular EMG datasets, Ninapro includes over 180 data acquisition sessions and is subdivided into 10 sub-datasets [2, 49]. We include some of the most popular sub-datasets used for benchmarking machine learning algorithms: Ninapro DB2, DB3, and DB5 [49, 11]. The Ninapro DB5 dataset uses two sets of a low-cost wireless wearable EMG device called the Myo Armband [49] worn on the same arm. Ninapro DB2 and DB3 use individually placed dry electrodes. All users in Ninapro DB3 have transradial amputations [2].

**CapgMyo**   Introduced in Geng et al. [50] and further described in Du et al. [6], the CapgMyo dataset includes 3 sub-datasets: DB-a, DB-b, and DB-c. We include DB-b in our benchmark dataset, which has multi-session data. In total, CapgMyo DB-b includes 10 subjects performing 8 gestures and data recorded by 128 high-density electrodes, separated into 8 acquisition modules with 16 electrodes per module.

**Myo Dataset**   Described in Côté-Allard et al. [7], the Myo dataset uses the wireless wearable EMG device called the Myo Armband. Like in [51], we include the evaluation dataset, which has 18 subjects. In this dataset, 7 different gestures are recorded and the methodology for placing the Myo Armband on the arm is specified in detail.

**EMG Data for Gestures Dataset**   The EMG Data for Gestures (UCI EMG) dataset [9], hosted on the UC Irvine Machine Learning Repository, has been used to benchmark leave-subject-out tests Lu et al. [18]. The device used is the Myo Armband. This dataset includes 36 subjects and includes 2 sessions, with only 1 gesture performance for each of 6 or 7 gestures performed per session. Although the timespan between the two sessions is not specified, both sessions for each participant are collected on the same day, as indicated by date-based timestamps in the filenames.

**Multi-channel sEMG Dataset**   The multi-channel sEMG (MCS) dataset presented in Ozdemir et al. [8] uses 4 bipolar Ag/AgCl electrode channels individually placed on 40 subjects' arms, placed by approximate locations of the specific muscles. Electrolyte gel is placed on the arm under the Ag/AgCl electrodes. The muscles measured from are the extensor carpi radialis, flexor carpi radialis, extensor carpi ulnaris, and flexor carpi ulnaris.

**Hyser Dataset**   The Hyser dataset, with details specified in Jiang et al. [10], includes data from 20 subjects and 34 gestures while recording from 256 gelled electrodes separated into 4 grids of high density electrodes, with two grids on the flexor side and two grids on the extensor side. Although there does not seem to be details on the exact materials used in the device, the most common gelled electrode materials used are Ag/AgCl along with a Cl electrolyte gel.

**FlexWear-HD Dataset**   We present a 13-person EMG dataset using a reusable electrode array called the FlexWear-HD dataset. This array uses a flexible printed circuit board (FPCB) with 64 hydrogel electrodes placed onto gold-plated copper pads on the FPCB. Two sessions are presented per subject, with the time between each session being about 1 hour. The wearable array is also kept on between the two sessions, allowing for the evaluation of typical changes that occur over time on EMG signals without the effects that can occur from replacing the electrode array on the arm.

# 4 Methods

This section outlines our methods for developing and evaluating gesture classification for EMG. We describe preprocessing techniques to convert time-series EMG data into 2D activity maps, the gesture classification models tested, and the generalization tasks for cross-subject and cross-session performance. Additionally, we detail the classification metrics and hardware setup used in our experiments. We note that we always use a constant numbers of epochs for training the classification models.

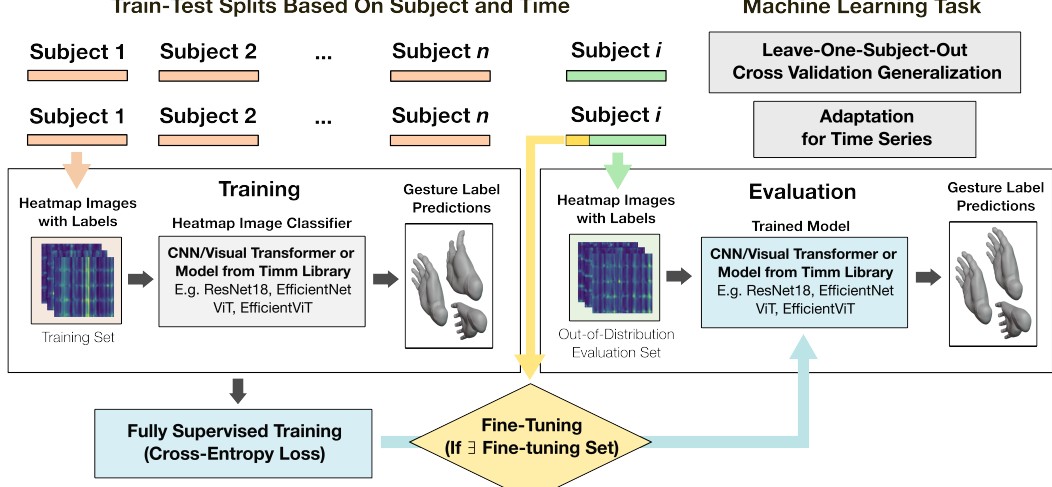

Figure 2: **Training pipeline for testing generalization and adaptation.** We show the training and evaluation pipeline for our benchmarking script to evaluate generalization across subjects and over time. For *leave-one-subject-out cross validation* (LOSO-CV), the training set involves all subjects other than the test subject, with the left-out subject $i$ changing from $1$ to $n$ between training runs. In addition, we test for *adaptation for time-series*, where data from the beginning of subject $i$ is used for fine-tuning after pretraining a model on data from other subjects. In our benchmark, both *few-shot fine-tuning* and *intersession fine-tuning* are used to evaluate adaptation for time-series.

## 4.1 Preprocessing Methods

A variety of preprocessing methods have been proposed for model training on EMG data. Given the time-series nature of EMG data, feature extraction methods include root-mean-square (RMS), number of zero-crossings, and mean absolute value [52]. By converting raw time series data or other manual features over time to heatmaps, we are able to create spatiotemporal patterns from time-series data, which can be given as input into 2D CNNs. We convert time series data for each electrode into separate rows in the activity map. Another approach to convert time-series data into a 2D format involves time-frequency transforms, such as spectrograms and continuous wavelet transforms (CWT). In this benchmark, we evaluate preprocessing using 1) heatmaps from raw data, 2) heatmaps from RMS windows, 3) spectrograms, and 4) CWTs. We show examples of these preprocessing methods resulting in activity maps in Figure 5.1. Classification of EMG data using 2D CNNs after preprocessing has achieved high accuracy in prior studies [50, 32]. We review ways that EMG has been processed into activity maps in Appendix A.3.

## 4.2 State-of-the-Art Image Classifier Algorithms

Several image classifier models have been successfully applied to EMG data [32, 50, 53, 54]. Following the examples of prior EMG classification work by Ozdemir et al. [32] and Dere and Lee [53], we evaluate on the ImageNet-pretrained ResNet18, and Visual Transformer (ViT) models [55, 56]. Additionally, we test the EfficientNet and EfficientViT models, which have shown strong performance on ImageNet classification while using fewer model parameters or achieving faster inference compared to other state-of-the-art visual models [57, 58]. The number of parameters used for each model is 2 million for EfficientViT, 4 million for EfficientNet, 6 million for ViT, and 11 million for Resnet18. By utilizing the PyTorch Image Models library in our benchmarking code, we enable other researchers to easily benchmark different visual classifier models by simply modifying the configuration file, with support for both pretrained and untrained models.

### 4.3 Different Generalization and Adaptation Tasks

Generalization and adaptation across different setups is essential for robust EMG-based models for real-world deployment. We evaluate two tasks: 1) generalization on a left-out subject, and 2) adaptation using initial data from a subject. For this second task, we fine-tune a pretrained model using initial data from the evaluation subject or the first session from the evaluation subject, respectively.

In task 1, we use LOSO-CV to assess how well an EMG gesture classifier and interface may work for a new user whose data was not included in training. In task 2, we perform TSTS on the data data to assess few-shot learning through fine-tuning, determining the minimal data required from a new subject for robust classification. We also test intersession accuracy to evaluate how well an EMG interface works for a user in a new session without recalibration. Train-test splits for generalization and adaptation are illustrated in Figure 3. More details on the classification metrics and data splits are provided in Appendix A.6.

### 4.4 Hardware Used for Benchmarking

In our benchmarking experiments, we use GPU nodes from the Pittsburgh Supercomputer Center, which have eight NVIDIA Tesla V100-32GB SXM2 GPUs each. Each node uses two Intel Xeon Gold 6248 "Cascade Lake" CPUs, and 512GB of DDR4-2933 RAM. In total, our tests took around 10,000 GPU hours. Much of the compute required comes from training individual models for each individual subject in order to evaluate average classification metrics across subjects.

## 5 Results

In this section, we present the outcomes of our experiments, highlighting the generalization and adaptation performance of gesture classification models using EMG. We evaluate the effectiveness of various data preprocessing techniques, followed by benchmarking different machine learning architectures. We then vary the amount of data used for fine-tuning, and present results on generalizing and adapting to data that comes from multiple sessions. The results provide insights into the generalization and adaptation tasks across different datasets, offering insight into the robustness and applicability of gesture classifiers in real-world scenarios.

### 5.1 Data Representation Benchmarking

Generalization and few-shot fine-tuning adaptation results for raw heatmap images, RMS heatmap images, spectrograms, and CWT preprocessing methods are summarized in Table 1. Each value represents the mean gesture recognition performance averaged over all $N$ models and train-test splits from LOSO-CV, where $N$ is the number of subjects in a given dataset. Leveraging insights from prior studies [32, 59], which highlighted the effectiveness of an ImageNet pre-trained ResNet model for gesture recognition, we adopted the ImageNet pre-trained ResNet-18 model to benchmark these preprocessing methods.

The experimental workflow involved pretraining the model using data from all training subjects, followed by fine-tuning with the first 20% of data from the left-out evaluation subject. Data splits were stratified by gesture, ensuring balanced representation across the fine-tuning process. A flowchart illustrating this process is provided in Figure 3. Fine-tuning significantly improved test accuracy across all four preprocessing methods, demonstrating the effectiveness of incorporating even small amounts of subject-specific data into the training process.

Performance varied across preprocessing methods and datasets. Raw heatmaps, spectrograms, and CWTs performed well overall, but their relative effectiveness depended on the dataset. For instance, time-frequency transforms significantly outperformed raw heatmaps for the MCS and Hyser datasets. Prior work has not comprehensively compared the performance of raw heatmaps and time-frequency transforms on the Hyser dataset, with Li et al. [17] focusing solely on raw heatmaps. Similarly, earlier studies on the MCS dataset [32] evaluated only CWT and STFT preprocessing, neglecting raw and RMS-based methods.

One notable gap in prior research is the exploration of phase information from time-frequency transforms, such as the STFT and Hilbert-Huang transform, for EMG-based gesture classification. Our findings, detailed in Appendix A.8, reveal that phase-based preprocessing methods underperform

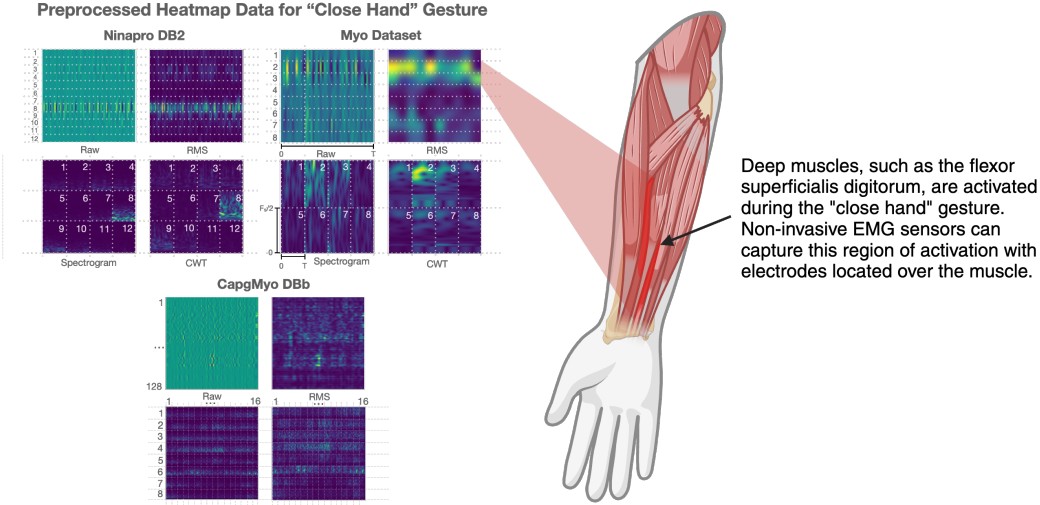

**Figure 3: Varying preprocessing methods for generating heatmaps.** Samples from different preprocessing methods are shown for the 128 electrode CapgMyo dataset, the 12 electrode NinaPro DB2, and the 8 electrode Myo Dataset. Values on the heatmaps correspond to the index of the electrode for the sub-image shown on the grid. All samples correspond to the closed-hand gesture. The closed-hand gesture primarily activates deep muscles of the flexor side of the forearm, such as the flexor digitorum superficialis.

| | Myo Dataset | UCI | NinaproDB5 | Capgmyo | NinaproDB2 | NinaproDB3 | MCS | Hyser | FlexWear-HD |
|---|---|---|---|---|---|---|---|---|---|
| **Pretraining before few-shot fine-tuning, LOSO-CV** | | | | | | | | | |
| Raw | **74.5/94.8** | 76.4/94.7 | **41.3/79.4** | 42.4/81.0 | 18.4/62.0 | 11.2/52.0 | 67.7/93.2 | 44.4/82.1 | **77.7/97.4** |
| RMS | 68.7/92.5 | 75.4/94.5 | 39.2/76.1 | 41.8/81.4 | 17.2/60.1 | 11.2/52.8 | 58.2/88.0 | 48.5/83.5 | 75.6/97.2 |
| STFT | 72.8/94.3 | **78.2/95.6** | 41.3/78.1 | 41.9/80.4 | 19.2/62.6 | **12.0/52.8** | 74.4/95.6 | 54.4/88.9 | 75.3/96.8 |
| CWT | 69.7/92.6 | 75.9/94.1 | 38.2/77.2 | **42.9/81.5** | **20.1/63.2** | 10.9/51.6 | **77.9/96.7** | **58.0/90.4** | 75.1/97.2 |
| **Finetuning with first 20% of data from left-out subject** | | | | | | | | | |
| Raw | **95.1/99.5** | 91.6/99.2 | **68.3/94.1** | **92.8/99.5** | 52.2/86.0 | 44.2/80.0 | 89.5/98.5 | 79.1/96.0 | 95.2/99.6 |
| RMS | 94.1/99.3 | 91.2/99.2 | 67.7/93.4 | 89.4/99.0 | 49.4/84.5 | 40.6/77.5 | 78.0/95.6 | 84.8/89.1 | 95.7/99.7 |
| STFT | 95.0/99.5 | **91.7/99.1** | 66.6/93.1 | 92.2/99.5 | **53.1/86.5** | **44.7/80.2** | 90.9/99.0 | 89.4/98.6 | **95.8/99.7** |
| CWT | 92.5/99.0 | **91.7/98.8** | 62.1/91.0 | 89.7/99.2 | 51.9/85.9 | 43.4/79.6 | **92.1/99.1** | 90.3/98.8 | 95.6/99.8 |

Table 1: **Benchmark of EMG Preprocessing Techniques.** LOSO-CV average test accuracy (Acc) / area under the receiver operating characteristic (AUROC) for all the datasets and for four common EMG-to-image preprocessing methods: original temporal EMG signals (Raw), root-mean-square (RMS), short-time fourier transform (STFT), and continuous wavelet transform (CWT), as mentioned in Section 4.1. RMS consists of taking consecutive windows of length 16-20 timesteps and applying the RMS transform.

compared to magnitude-based approaches like STFT or CWT. This performance disparity may be attributed to the absence of magnitude information, which effectively captures variations in EMG activity across electrodes, as illustrated in Figure 5.1. For all experiments reported in Table 1, models were pretrained over 100 epochs and fine-tuned over 750 epochs.

## 5.2 Machine Learning Architecture Benchmarking

Evaluating diverse machine learning architectures is crucial for identifying the most effective models for EMG gesture classification, particularly when using heatmaps, spectrograms, or CWTs. We assess both convolutional neural networks (CNNs) and vision transformers (ViTs) due to their distinct strengths: CNNs leverage convolutional and pooling layers to learn locally shift-invariant features [55], while transformers utilize self-attention mechanisms to capture complex spatial relationships across entire images [56]. Our benchmarks include architectures known for strong performance in EMG classification [32, 53, 59], specifically CNNs [8, 59, 50] and ViTs [53], pretrained on the ImageNet

dataset [60] with all weights unfrozen during training. Pretraining is conducted over 50 epochs, followed by fine-tuning over 375 epochs.

For each dataset, we select the best-performing preprocessing method from Table 1 (highlighted in bold) and evaluate leave-one-subject-out cross-validation (LOSO-CV) and few-shot transfer learning using four image classifiers: 1) ResNet18, 2) EfficientNet, 3) ViT, and 4) EfficientViT. All models are initialized from an ImageNet-1k pretrained baseline [60].

Results reveal that ResNet18 achieves the best performance on Ninapro DB5, Ninapro DB2, and MCS. EfficientNet outperforms others on CapgMyo, UCI, Ninapro DB3, and FlexWear-HD, while EfficientViT performs best on the Myo Dataset and Hyser. Notably, EfficientNet and EfficientViT have not previously been applied to EMG signal classification but are recognized for their state-of-the-art performance and inference speed in computer vision tasks [58, 57].

Tables 1 and 2 indicate that datasets utilizing array-based wearable electrode devices, such as the Myo Dataset, UCI EMG dataset, and FlexWear-HD, generally exhibit superior performance in LOSO-CV compared to datasets requiring individually placed electrodes or uniform electrode grids, as seen in CapgMyo and Hyser. Exceptions include the Ninapro DB5 and MCS datasets. For Ninapro DB5, despite its use of two Myo Armband devices, classification performance is lower, potentially due to noisy labels [61], which may artificially suppress generalization performance. Conversely, the MCS dataset achieves relatively high LOSO-CV performance despite having the fewest electrodes. This may be attributed to its electrode placement methodology, where expert researchers position individual electrodes on a few specific muscles, potentially reducing concept shifts across subjects [8]. However, this approach is challenging to replicate for end-users who must independently don the devices. Additionally, it significantly increases the donning time compared to using a single, integrated device.

| | Myo Dataset | UCI | NinaproDB5 | Capgmyo | NinaproDB2 | NinaproDB3 | MCS | Hyser | FlexWear-HD |
|---|---|---|---|---|---|---|---|---|---|
| **Before few-shot finetuning through pretraining using leave-one-subject-out cross validation** | | | | | | | | | |
| RN18 | 73.7/94.9 | 76.6/95.0 | **42.1/78.9** | 41.4/82.9 | **19.9/63.0** | 11.1/51.6 | **77.8/96.6** | 56.6/89.1 | 77.0/97.4 |
| EN | 73.5/94.2 | **78.6/95.3** | 39.9/78.6 | **46.6/84.2** | 19.5/63.0 | **11.6/52.3** | 76.9/96.4 | 58.6/90.7 | **83.9/98.5** |
| ViT | 70.8/93.2 | 77.3/95.1 | 38.8/77.7 | 39.8/78.9 | 18.0/61.1 | 11.1/51.3 | 76.3/96.0 | 56.3/88.8 | 70.2/96.3 |
| EViT | **73.8/93.9** | 77.2/94.4 | 40.9/78.7 | 45.0/82.8 | 18.8/62.0 | 11.1/51.3 | 76.2/96.2 | **61.8/92.2** | 74.9/96.3 |
| **After few-shot fine-tuning by doing training using first 20% of data from left-out subject** | | | | | | | | | |
| RN18 | 94.6/99.5 | 91.4/99.0 | 68.5/93.6 | **91.2/99.3** | 52.4/85.8 | **45.2/81.3** | 91.0/99.0 | **87.4/98.4** | 95.9/99.8 |
| EN | **94.9/99.6** | **91.7/99.1** | 67.7/93.3 | 89.7/98.6 | 53.8/87.1 | 45.1/81.0 | 91.0/99.0 | 85.4/97.8 | **96.9/99.8** |
| ViT | 94.9/99.4 | 90.1/98.1 | 68.2/94.2 | 85.1/98.5 | 51.0/86.1 | 41.2/78.5 | 91.4/98.9 | 70.6/93.3 | 93.6/99.5 |
| EViT | 94.9/99.4 | **91.7/99.1** | **69.2/94.1** | 87.1/98.3 | **54.0/86.9** | 44.8/81.4 | **92.0/98.8** | 83.2/96.6 | 95.3/99.6 |

Table 2: **Benchmarking of Machine Learning Architectures.** Performance of gesture recognition models on each dataset (Acc/AUROC). The models used are the ImageNet-pretrained ResNet18 (RN18), EfficientNet (EN), ViT, and EfficientViT (EViT).

## 5.3 Varying Amount of Data Available for Training

Using the best-performing preprocessing method identified in Table 1 and the corresponding best classifier architecture from Table 2 for each dataset, we evaluate adaptation methods, as presented in Table 3. For the FT-$X\%$ rows, fine-tuning is performed using the first $X\%$ of data from the left-out subject after pretraining the model on data from all other subjects. Gesture stratification ensures a balanced fine-tuning set. By varying the amount of adaptation data, we assess the data requirements for achieving significant performance gains.

Our findings indicate that performance generally improves with more fine-tuning data from the left-out subject. Notably, for the Myo Dataset and FlexWear-HD, performance reaches 93.1% for 7 gestures and 94.2% for 10 gestures, respectively, with just 5% of the subject's initial data. This corresponds to approximately eight seconds of data for the Myo Dataset and 30 seconds for FlexWear-HD. These results highlight that minimal data collection from a new subject is sufficient for achieving high classification accuracy when fine-tuning a model pretrained on a large dataset of training subjects. On average, accuracy improves by 44% compared to the pretrained model after fine-tuning with just 5% of the subject's data. This aligns with findings from Sussillo et al. [14], who reported a 30% performance improvement on EMG tasks after model personalization via fine-tuning. For Table 3,

all experiments are repeated with three seeds, and we report the mean and standard deviation of test accuracies, observing minimal variance in performance metrics.

For intersession tests shown in Table 4, we pretrain using data from all available sessions for the training subjects and fine-tune with data from the first session of the left-out subject. This differs from the evaluations in Tables 1, 2, and 3, where only the first session is used for training, fine-tuning, and evaluation. Notably, only four datasets are included in Table 4 because they contain data from more than two sessions. For datasets like CapgMyo and Hyser, where sessions are recorded on different days, fine-tuning performance is significantly lower than in Table 3, where fine-tuning data is from the same day as the test data.

For datasets with transition data, Table 5 shows performance when using data windows that include both isometric holds and transitions between gestures. As expected, these datasets exhibit lower accuracies compared to Table 3 due to the increased variability in EMG signals introduced by dynamic transitions. All pretraining experiments are conducted over 50 epochs, and fine-tuning is performed over 375 epochs.

| Task | Myo Dataset | UCI EMG | Ninapro DB5 | Capgmyo | Ninapro DB2 |
|------|-------------|---------|-------------|---------|-------------|
| FT-5% | 93.2/99.3 (0.1/0.1) | 82.1/96.7 (0.8/0.3) | 48.4/83.7 (0.6/0.3) | 76.1/95.9 (2.2/0.1) | 37.1/76.4 (0.4/0.3) |
| FT-20% | 95.0/99.4 (0.3/0.1) | 91.7/99.1 (0.5/0.1) | 69.7/94.1 (1.1/0.4) | 89.2/98.7 (0.9/0.2) | 52.3/86.0 (0.4/0.2) |
| FT-40% | 98.3/99.9 (0.1/0.0) | 92.5/99.3 (0.3/0.1) | 76.0/96.4 (0.6/0.2) | 96.1/99.7 (0.3/0.1) | 55.9/88.8 (0.3/0.2) |
| FT-60% | 98.9/100.0 (0.1/0.1) | 95.0/99.6 (0.3/0.1) | 82.0/97.8 (0.1/0.1) | 96.4/99.9 (0.4/0.2) | 56.4/89.1 (0.7/0.2) |
| FT-80% | 99.6/100.0 (0.1/0.0) | 97.4/99.8 (0.9/0.1) | 78.1/96.9 (0.6/0.3) | 97.7/99.9 (0.6/0.1) | 56.0/90.1 (0.7/0.3) |

| Task | Ninapro DB3 | MCS | Hyser | FlexWear-HD |
|------|-------------|-----|-------|-------------|
| FT-5% | 32.7/72.0 (0.5/0.7) | 81.1/96.5 (0.9/0.3) | 66.5/91.1 (2.8/1.3) | 94.6/99.6 (0.7/0.1) |
| FT-20% | 44.4/80.3 (0.3/0.2) | 91.4/99.0 (0.7/0.1) | 84.1/96.5 (1.3/0.2) | 96.8/98.8 (0.2/1.6) |
| FT-40% | 49.6/83.2 (0.3/0.2) | 95.6/99.7 (0.1/0.0) | 91.2/98.3 (1.7/0.5) | 98.9/99.9 (0.1/0.1) |
| FT-60% | 51.2/85.1 (0.8/0.4) | 96.6/99.8 (0.3/0.1) | 92.0/98.5 (0.7/0.5) | 98.8/99.8 (0.3/0.0) |
| FT-80% | 52.7/87.3 (1.1/0.5) | 97.2/99.8 (0.2/0.1) | 96.7/99.0 (0.4/0.4) | 99.3/100.0 (0.2/0.0) |

Table 3: **Benchmarking Amount of Data for Fine-tuning.** Performance on datasets for adaptation tasks useful in EMG control interfaces (Mean Acc/Mean AUROC with standard devations in parentheses). LOSO-CV stands for leave-one-subject-out cross validation. IS FT stands for intersession fine tuning. FT-X% involves using the first X% of data from the left-out subject for fine-tuning after pretraining with data from all others subjects.

| Task | UCI | Capgmyo | Hyser | FlexWear-HD |
|------|-----|---------|-------|-------------|
| IS w/o FT | 79.8/95.5 | 47.9/86.4 | 72.5/95.5 | 81.7/97.6 |
| IS FT | 93.5/99.6 | 62.1/91.4 | 60.1/91.2 | 99.1/100.0 |

Table 4: **Performance on datasets for generalization and adaptation across sessions, or intersession performance.** In this case, the test set is the second session of a left-out subject. FT stands for fine-tuning. IS FT stands for intersession fine tuning. Before fine-tuning, the model is trained on all subjects other than the left-out subject. Fine-tuning involves training on the left-out subject's first session.

| Task | UCI EMG | Ninapro DB5 | Ninapro DB2 | Ninapro DB3 | MCS |
|------|---------|-------------|-------------|-------------|-----|
| **Varying proportions, transitions classified, finetuning** | | | | | |
| FT-20% | 93.0/99.1 | 59.6/91.0 | 47.6/84.4 | 45.5/80.8 | 85.3/97.6 |
| FT-40% | 93.6/99.5 | 66.5/93.5 | 48.2/84.7 | 45.8/80.9 | 88.0/98.5 |
| FT-60% | 95.0/99.5 | 72.5/95.4 | 46.6/84.1 | 44.2/79.5 | 91.0/99.1 |
| FT-80% | 98.0/99.9 | 65.0/93.1 | 44.7/82.5 | 40.3/78.2 | 89.0/98.7 |

Table 5: **Performance when including transition data.** This table shows performance when including transition data and not only during isometric holds of a gesture. The label on the entire window of data is set as the label of the data at the last time step of the window.

### 5.4 Domain Generalization Algorithms

We test additional training algorithms, which have been tested in previous work on domain generalization, to compare how well they work compared to standard supervised learning [62, 63]. In these experiments, we again use the best performing preprocessing method and model architecture for the dataset, but include the use of invariant risk minimization (IRM), and correlation alignment (CORAL) [62, 63]. We test these algorithms that take into account the different domains during training: 1) IRM minimizes a loss that penalizes models where the optimal classifier differs across domains, and 2) CORAL aligns the covariances of the feature representations between domains through a loss term. The resulting generalization and few-shot fine-tuning results are shown in Appendix A.9. Overall performance is comparable to with training using standard cross-entropy loss, which is a similar result found in Gulrajani and Lopez-Paz [62] and Koh et al. [63].

## 6    Limitations and Future Work

The current benchmarking studies in this work have not evaluated EMG classification performance with activity maps exceeding a resolution of 224x224 pixels. Higher resolution activity maps may enhance inter-subject performance when employing spectrogram or continuous wavelet transform (CWT) image preprocessing techniques, particularly for datasets with a greater number of electrodes. This is due to the potential reduction in downsampling of the resulting time-frequency transformed images for each electrode, thereby preserving specific features within the transformed data.

Among the datasets analyzed (Table 2), the FlexWear-HD dataset achieved the highest LOSO-CV accuracy, registering an 83.9% test accuracy for 10-class classification. While robust generalization to new subjects remains a significant challenge for EMG-based control interfaces [14], future work leveraging larger datasets and models holds promise for developing a universal EMG classifier. This classifier could rapidly generalize or adapt to EMG data from new subjects, enabling a more reliable deployment of EMG-based control systems.

Datasets such as CapgMyo, Hyser, Myo Dataset, and FlexWear-HD do not present transition data between gestures, likely due to the complexity of labeling short, dynamic transitions in gesture-based classification systems. Notably, in prior work with FlexWear-HD [1], users successfully controlled robots in real time using classified EMG data streams, even without transition data in the training process. For datasets that do include transition data, such as Ninapro, MCS, and UCI EMG, Table 5 presents results where transition windows are also classified.

Currently, none of these datasets capture gesture data performed during real-world activities, where users interact with computers or robots outside controlled, cue-based environments. Developing a dataset with ground-truth labels for spontaneously performed gestures in natural settings would address this gap. Such a dataset could better represent the variability of real-world gestures, ultimately enhancing the adaptability and performance of EMG-based gesture classification systems.

## 7    Conclusion

This study introduces a new tool for benchmarking EMG datasets, providing insights into the effectiveness of various classification methods, EMG preprocessing techniques, potential for real-world applications, and open research problems in learning-based EMG gesture classification for the research community. The performance of these methods shows promising results for enhancing the generalization of EMG-based systems, especially through a standardized format to compare performances metrics. While benchmarking, we find that adaptation using data from the validation subject can significantly enhance performance, requiring only a small amount of data from the subject.

## Acknowledgments

This material is based upon work supported by the National Science Foundation under Grant No. 2341352 and Grant No. DGE2140739.

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

# A Appendix / supplemental material

## A.1 EMG Signals

EMG sensors detect voltage signals due to motor neurons activating the attempted contraction of muscle fibers. Muscle contraction within the body is caused by the activation of motor neurons that innervate a set of around 100 to 1000 muscle fibers. This set of innervated muscle fibers is called a motor unit [19]. When a neuron's voltage spike propagates to the end of its axon, the neuron releases acetylcholine neurotransmitters into the neuromuscular junction, activating acetylcholine-modulated ion channels that allow sodium ions to enter the muscle cell. The influx of sodium ions causes voltage-modulated ion channels for sodium to open as well, propagating the voltage spike from the action potential along the muscle fiber [64].

Once the sodium voltage spike propagates along the muscle fiber membrane and reaches an area of the membrane called the T-tubules, a voltage-sensitive reaction is triggered, causing calcium ion channels in the sarcoplasmic reticulum organelle to open. When calcium ions move into the cytoplasm of the muscle fiber, they bind to troponin proteins within the muscle fiber, exposing binding sites on actin filaments. Myosin heads bind to these sites and pull actin filaments toward the middle of a sarcomere, causing muscular contraction [64].

Because of the speed of the propagation of the action potential in a muscle fiber and the superposition of voltage signals of a population of motor units firing at once [65], examinations involving surface EMG will see signals of interest from EMG activity ranging from 10 Hz to 500 Hz [66]. This distribution of EMG activity over frequency is dependent on both impedances in the body between the electrode and the muscle fibers, and on the proportion of slow-twitch (Type I) and fast-twitch (Type II) muscle fibers in the anatomy [64]. The spatial distribution of EMG activity around the arm further distinguishes the specific intended gesture of the subject [65]. For example, the activation of more muscle fibers on the palmar side of the forearm (specifically the flexor carpi muscles) may distinguish wrist flexion from other gestures. The activation of different proportions of slow and fast muscle fibers as well as the spatial distributions of muscle activation change over time depending on the amount of fatigue and amount of effort attempted by the user, for example due to muscular compensation [43, 42]. By detecting voltage signals from patterns of muscle voltage spikes over time and spatially around the forearm, we are able to decipher high-dimensional motor intent from as many as 20 muscles around the forearm.

## A.2 EMG Devices

A common EMG device, used by 3 of the datasets in our benchmark, is the Myo Armband. The Ninapro DB5, Myo Dataset, and the UCI EMG dataset all use this device. Each armband contains 8 stainless steel electrodes. Due to the use of dry stainless steel electrodes without the use of electrolyte gel and in order to save energy for a battery-powered wireless wearable device, respectively, these devices have relatively high electrode-skin impedances [67] and relatively low sampling rates (200 Hz).

Other common devices used for EMG measurements are the individually-placed bipolar Delsys Trigno sensors, which use dry silver electrodes [68]; high-density EMG arrays, which use gels on top of exposed metal pads on flexible printed circuit boards [50]; as well as individually-placed sticky hydrogel electrodes, which are often commonly used in electrocardiograms [8].

## A.3 EMG Preprocessing into Heatmaps

Preprocessing EMG time-series data as heatmap images and classifying them using CNNs has yielded high classification accuracy results in prior work [50, 32], setting the state-of-the-art for gesture classification in 2016 [50] for some popular EMG datasets, including Ninapro DB1, Ninapro DB2, and the CSL-HDEMG. After this work, many papers used similar heatmap image preprocessing methods, particularly by transforming raw data into heatmaps [69], first doing feature extraction methods such as root-mean-square windows before transforming into heatmaps [1], or first transforming the images into time-frequency plots (such as spectrograms, or continuous wavelet transforms) [32].

| Dataset | Channels | Subjects | Gestures | Sample Len (ms) | Step Len (ms) | Sampling Freq (Hz) | Sessions | Samples |
|---|---|---|---|---|---|---|---|---|
| Myo Dataset | 8 dry | 18 | 7 | 250 | 50 | 200 | 1 | 48030 |
| UCI EMG | 8 dry | 36 | 6 | 250 | 50 | 1000 | 2 | 12870 |
| NinaproDB5 | 16 dry | 10 | 10 | 250 | 50 | 200 | 1 | 39597 |
| Capgmyo | 128 gelled | 10 | 8 | 250 | 50 | 1000 | 2 | 25600 |
| NinaproDB2 | 12 dry bipolar | 40 | 10 | 250 | 250 | 2000 | 1 | 38810 |
| NinaproDB3 | 12 dry bipolar | 11 | 10 | 250 | 50 | 2000 | 1 | 64426 |
| MCS | 4 gelled bipolar | 40 | 7 | 250 | 250 | 2000 | 1 | 28000 |
| Hyser | 256 gelled | 20 | 10 | 250 | 125 | 2048 | 2 | 8239 |
| FlexWear-HD | 64 hydrogel | 13 | 10 | 250 | 50 | 4000 | 2 | 46116 |

Table 6: Summary of information on EMG datasets. The number of sample lengths, and step lengths were prescribed in this study. The number of gestures used is also less than all that is available for the UCI EMG, Hyser, NinaproDB2, NinaproDB5, and MCS datasets similar to other studies that tested generalization [18, 32, 17]. The number of reported samples are from session 1, as all studies other than those in Table 4 use only the first session.

## A.4 Dataset Details in Benchmark

In Table 6, we show details about the datasets in this benchmark in a condensed table format. For the UCI EMG, Hyser, NinaproDB2, NinaproDB5, and MCS datasets, we use less than the total number of gestures available, similar to other studies that have tested generalization; we select these gestures based on papers that have previously tested with these datasets before and that are based on basic wrist and finger movements [18, 32, 17]. For most datasets, we use a step length of 50 ms, although for some datasets with a high number of electrodes and sampling rate, we increase the step length to make computation more tractable.

| Dataset | Rest | Radial | Flexion | Ulnar | Extension | Fist | Abduction | Adduction | Supination | Pronation |
|---|---|---|---|---|---|---|---|---|---|---|
| Myo Dataset | 6861 | 6861 | 6855 | 6864 | 6863 | 6865 | 6861 | | | |
| UCI EMG | 4286 | 4325 | 4273 | 4347 | 4310 | 4140 | | | | |
| NinaproDB5 | 4102 | 3335 | 4199 | 4756 | 3709 | 3614 | 3545 | 3926 | 4326 | 4085 |
| Capgmyo | | | | | | 1600 | 1600 | 1600 | | |
| NinaproDB2 | 4183 | 3971 | 3149 | 3863 | 3091 | 3555 | 4094 | 4090 | 4456 | 4358 |
| NinaproDB3 | 6556 | 6511 | 6123 | 6341 | 6085 | 6206 | 6477 | 7075 | 7245 | 5807 |
| MCS | 4000 | 4000 | 4000 | 4000 | 4000 | 4000 | 4000 | | | |
| Hyser | | 1582 | 1680 | 1673 | 1666 | 1680 | 1680 | | 1638 | 1638 |
| FlexWear-HD | 4644 | 4572 | 4572 | 4572 | 4680 | 4644 | 4644 | | 4644 | 4644 |

Table 7: The number of samples for common gestures in each EMG dataset. The full gesture names are Radial and Ulnar Deviation; Finger Abduction and Adduction; Wrist Flexion, Extension, Supination, and Pronation.

We specify the number of samples for each gesture in Tables 7 and 8. All samples were created from overlapping 250 ms windows of the raw data, where each dataset's step length determined the amount of time between the start of one window and the next. 250 ms was selected as the standard sample length since longer window lengths generally improve classification but a delay of over 250 ms would make the classifier unsuitable for real-time usage [70]. Windows at the transition between gestures were excluded, ensuring that every window corresponds to a single gesture.

To prevent large class imbalance in Ninapro DB5, the rest gesture was subsampled to the average number of samples of the other gestures. The variation in sample numbers across a dataset is due to small differences in repetition length during data collection and incorrectly performed repetitions being withheld from the dataset.

| Dataset | Gesture Type | Thumb, Index | Thumb, Middle | Thumb, Index, Middle | Thumb | Index | Index, Middle | All But Thumb |
|---|---|---|---|---|---|---|---|---|
| Capgmyo | Extension | – | – | 1600 | 1600 | 1600 | 1600 | 1600 |
| Hyser | Pinch | 1673 | 1603 | – | – | – | – | – |
| FlexWear-HD | Pinch | – | – | 4500 | – | – | – | – |

Table 8: The number of samples for less common gestures in each EMG dataset. The column names refer to the fingers involved in each gesture.

| | Myo Dataset | UCI EMG | NinaproDB5 | Capgmyo | NinaproDB2 | NinaproDB3 | MCS | Hyser | FlexWear-HD |
|---|---|---|---|---|---|---|---|---|---|
| Time per Repetition | 5 | 3 | 5 | 3 | 5 | 5 | 6 | 1 | 2 |
| Inter-Repetition Rest Time | 5 | 3 | 3 | 7 | 3 | 3 | 4 | 2 | 2.5 |
| Repetitions per Gesture | 4 | 1/1 | 6 | 10/10 | 6 | 6 | 5 | 3/3 | 10/5 |

Table 9: The number of seconds each repetition of a gesture was performed for, the number of seconds participants rested between repetitions, and the number of repetitions that were performed for each gesture. A / separates the repetitions in the first session from the second for datasets with multiple sessions.

| Reference Study | Chronological Split | Random Split | Mixed Split | Inter-Subject | Inter-Session | Inter-Trial |
|---|---|---|---|---|---|---|
| Zhang et al. [38] | ✗ | ✗ | ✓ | ✗ | ✗ | ✓ |
| Lin et al. [71] | ✗ | ✗ | ✓ | ✗ | ✗ | ✓ |
| Lee et al. [72] | ✗ | ✓ | ✗ | ✗ | ✗ | ✗ |
| Tuncer et al. [73] | ✗ | ✓ | ✗ | ✗ | ✗ | ✗ |
| Azhiri et al. [74] | ✓ | ✗ | ✗ | ✗ | ✗ | ✓ |
| Sri-Iesaranusorn et al. [37] | ✗ | ✗ | ✓ | ✗ | ✗ | ✓ |
| Fatimah et al. [39] | ✗ | ✓ | ✗ | ✗ | ✗ | ✗ |
| Chen et al. [75] | ✗ | ✗ | ✓ | ✗ | ✗ | ✓ |
| Rahimian et al. [76] | ✗ | ✗ | ✓ | ✗ | ✗ | ✓ |
| Kim et al. [77] | ✗ | ✗ | ✓ | ✗ | ✓ | ✗ |
| Fu et al. [78] | ✗ | ✓ | ✗ | ✗ | ✗ | ✗ |
| Abbaspour et al. [79] | ✗ | ✓ | ✗ | ✗ | ✗ | ✗ |
| Gautam et al. [80] | ✗ | ✓ | ✗ | ✗ | ✗ | ✗ |
| Ozdemir et al. [81] | ✗ | ✓ | ✗ | ✗ | ✗ | ✗ |
| He and Jiang [40] | ✗ | ✓ | ✗ | ✗ | ✗ | ✗ |
| Shen et al. [82] | ✓ | ✗ | ✗ | ✗ | ✗ | ✓ |
| Qi et al. [83] | ✓ | ✗ | ✗ | ✗ | ✓ | ✗ |
| Bhagwat and Mukherji [84] | ✗ | ✓ | ✗ | ✗ | ✗ | ✗ |
| Kim et al. [41] | ✗ | ✗ | ✓ | ✗ | ✓ | ✗ |
| Purushothaman and Vikas [85] | ✗ | ✓ | ✗ | ✗ | ✗ | ✗ |
| Saikia et al. [86] | ✗ | ✓ | ✗ | ✗ | ✗ | ✗ |
| Baldacchino et al. [87] | ✗ | ✗ | ✓ | ✗ | ✗ | ✓ |
| Phukpattaranont et al. [88] | ✗ | ✓ | ✗ | ✗ | ✗ | ✗ |
| Sezgin [89] | ✓ | ✓ | ✗ | ✗ | ✗ | ✗ |
| Jiralerspong et al. [90] | ✓ | ✗ | ✗ | ✗ | ✓ | ✗ |
| Rahimi et al. [91] | ✗ | ✓ | ✗ | ✗ | ✗ | ✗ |
| Ariyanto et al. [92] | ✗ | ✗ | ✓ | ✗ | ✗ | ✓ |
| Gijsberts et al. [93] | ✗ | ✗ | ✓ | ✗ | ✗ | ✓ |
| Al-Timemy et al. [94] | ✗ | ✗ | ✓ | ✗ | ✗ | ✓ |
| Khushaba et al. [95] | ✗ | ✗ | ✓ | ✗ | ✗ | ✓ |
| Tang et al. [96] | ✗ | ✗ | ✓ | ✗ | ✗ | ✓ |
| Khushaba et al. [97] | ✗ | ✗ | ✓ | ✗ | ✗ | ✓ |
| Naik and Kumar [98] | ✗ | ✗ | ✓ | ✗ | ✗ | ✓ |

Table 10: This table shows the ways in which each reference study splits its data into train and test sets. Chronological split has the test set as the last portion of the dataset to be collected. Random split has a random subset of the dataset as the test set; k-fold CV is included here. Mixed split has the test set consisting of distinct subjects, sessions, or trials not collected last chronologically. Inter-subject, inter-session, and inter-trial refer to the dataset attribute that chronological splits and mixed splits partitioned across.

In this table we show the data splitting methods used by all 33 EMG gesture classification studies reviewed in [11]. Studies were systematically selected based on several criteria including being recent and published after 2010, being open access, and having finger gestures within the set of gestures for classification. We note that data splitting methods used in our study (chronological splitting, inter-subject, and inter-session) are much less common in the literature, but are more indicative of capacity for generalization than more commonly used methods. Notably, inter-session testing leads to more distribution shift than inter-trial since it usually involves the subject removing and reattaching the measurement device after a significant rest period; inter-subject testing has even greater distribution shift.

### A.5 Learning-based generalization or adaptation prior work

A small proportion of EMG classification papers use splits based on leave-one-subject-out (LOSO) cross validation, despite its importance for real-world deployment. None of the papers reviewed in the EMG classification review by Sultana et al. [11] evaluate using LOSO-CV or other intersubject accuracy tests. Among the works that do test on intersubject accuracy, Ozdemir et al. [32] presents a fine-tuned ResNet50 model on a 4-electrode dataset, achieving a 94.41% LOSO-CV accuracy for

7 gestures. Similarly, Li et al. [17] reports CNN models that attain an 85.4% LOSO-CV accuracy using a dataset they collected with 256 electrodes for 10 gestures, although the published dataset uses different subjects. Additionally, works by Wang et al. [33], Zhang et al. [34], and Xu et al. [35] on their own unpublished datasets achieve LOSO-CV accuracies of 80.3% for 5 gestures, 73% for 6 gestures, and 60.8% for 17 gestures, respectively.

In the literature, among the prior work that tests on leaving at least one subject out for a pre-existing EMG dataset published by another group, only one study has achieved greater than 50% accuracy. Lu et al. [18] achieves an average accuracy of 77% for 6 gestures. Instead of the typical LOSO cross validation, Lu et al. [18] uses a train-test split, randomly leaving nine out of thirty-six subjects as the test set, and utilizes an EMG dataset published by Krilova et al. [9]. Other studies by Islam et al. [29], Du et al. [6], and Wei et al. [30] that employ LOSO cross validation on previously available EMG datasets, such as NinaPro or CapgMyo, achieve approximately or less than 50% accuracy, with Islam et al. [29] and Wei et al. [30] not reporting their exact accuracies due to low performance.

There is a larger body of prior work that test on out-of-domain adaptation tasks, involving using small amounts of data, either labeled, unlabeled, or both, from the subject that the model is tested on. For example, work from Côté-Allard et al. [7], Li et al. [17], Zhang et al. [34], Islam et al. [29], Du et al. [6], and Wei et al. [30] all evaluate adaptation. All of this work varies in how much data is used, whether unlabeled or labeled data is used from the validation subject, and the datasets tested. A common version of these out-of-domain adaptation tasks involves **intersession accuracy tests (IAT)**, in which an entire session of data collection is left-out as the test set. This approach is practical for predicting gestures in a new session for a subject with data from a previous session. The test session can range from data collected on the same day [1, 9] to several days after the initial session [6, 17], and it varies based on whether the sensors were re-donned for the test session [6, 17] or continuously worn since the previous data collection [1].

### A.6 Classification Metrics

For all tests, we split the data into training, validation, and test sets. The training set includes data from all subjects except the left-out subject. The validation and test sets are evenly split subsets of the left-out subject's data, with the validation set being the earlier subset and the test set being the later subset in time. The test set is only evaluated at the final epoch. When evaluating task 2, we use a fine-tuning set that consists of an initial subset of data from the evaluation subject.

We report two classification metrics: average test classification accuracy and average area under the receiver operating characteristic (AUROC) for the test set in an average one-vs-rest split. Since our datasets are balanced, as shown in Table 8, the average test classification accuracy is unbiased towards the performance of any specific class. The AUROC is also commonly used to evaluate the classifier's discrimination ability.

### A.7 Additional Details on Datasets and Benchmarking

Although there are several published EMG datasets, there are several variables that can differ from dataset to dataset, for example: number of electrodes, electrode materials, recording hardware involving differing hardware amplifiers, hardware filters, and analog-to-digital converters, numbers of participants, sets of gestures, and sampling rates [2, 6–10]. Because there is no standard set of hardware or gestures that people have used for studies involving EMG gesture classification, it is important to find learning-based models that perform well across these different variables.

### A.7.1 Ninapro

From prior work, we note that Wei et al. [30] mentioned low leave-one-subject-out cross validation accuracy, reporting around 30% in preliminary experiments on the Ninapro DB1 dataset. This may be due to the difficulty required in precise individual sensor placements around the arm for use in gesture classification for the Ninapro DB1 dataset [68]. In addition, an assessment of the Ninapro datasets by Chang et al. [61] of DB2 to DB8 have found low signal to noise ratios for some sub-datasets, as well as occurrences of mislabeling across all these sub-datasets. The numbers of subjects, number of gestures, and the types of sensor can vary between the datasets [68, 2, 49], although the range of number of electrodes varies from only 10 to 16 electrodes.

### A.7.2 CapgMyo

From prior work by Du et al. [6], LOSO classification accuracy was reported to be 39.0%, although this rises to 55.3% using an adaptive batch normalization method when including data from the left-out subject. We note that the device used for measuring EMG seems to require the placement of 8 individual modules on the arm, which may cause significant variations required for individual placements between sessions and subjects. The minimum number of days between recording sessions is 7 days for the same subject.

### A.7.3 Myo Dataset

Prior work from Côté-Allard et al. [7] and Lin et al. [51] classifying the Myo dataset focuses on domain adaptation, using some data from the validation subject during training. These works show 98.31% and 94.53% classification accuracy, respectively. Although the former work uses the full 36 subject dataset, the latter exclusively uses the 17 subjects designated as the "evaluation dataset". Our benchmark also uses the evaluation dataset, which now includes 18 subjects. For this dataset, there are additional details on the placement of the Myo Armband on the arm, namely that orientation of the Myo Armband is placed such that the light on the armband faces the hand, the tightness of the armband is configured to maximum tightness, and the armband is slid onto the arm until the inner circumference of the armband matches the forearm [7]. Due to this placement procedure, based on figures included in [7], the Myo Armband is placed anywhere from the thickest part of the forearm (close to the elbow), to a few centimeters from the wrist.

### A.7.4 UCI EMG

Although there is not a significant amount of detail published with this dataset, it is specified that the Myo Armband device is used, and bluetooth is used to send to a PC from the device. In a test involving testing on left-out-subjects in Lu et al. [18], an average validation accuracy of 77% is achieved while leaving 9 out of 36 subjects out as the validation set at a time. We note that although the raw data is recorded at 1000 Hz while the Myo Armband can only record at 200 Hz, the authors seem to have upsampled the data. This can be seen in the raw data from the many repeated values across timesteps. We note that for Table 3, the UCI EMG dataset is fine-tuned on the first X% of samples after concatenating samples from both sessions together, where each session only has 1 gesture repetition

### A.7.5 Hyser Dataset

Similar to work from Li et al. [17], which achieves a LOSO-CV accuracy of 85.4% and a leave-one-session-out accuracy of 82.2% for a dataset collected using the same protocol as the Hyser dataset, in our benchmarking script, we also use only 10 of the 34 gestures for a more tractable learning problem for real-world use. We note we were not able to achieve the same LOSO-CV accuracy, although the subjects in the published dataset are different than the subjects in Li et al. [17]. We note that because Hyser subject 5 seems to only have the first 9 out of 10 gestures we classify for in their first session, we ignore the classification of Hyser subject 5 for the results tables we generate because we want to keep a consistent number of gestures between subjects for classification. Data collection sessions are on separate days for the same subject.

### A.7.6 FlexWear-HD Dataset

The FlexWear-HD dataset includes 13 subjects, who perform 15 total repetitions of 10 gestures. An easy-to-wear, reusable high density 64-electrode hydrogel array is used, with the device wrapped around the proximal forearm with a Velcro strip. The strip is placed approximately in the same orientation across subjects with palpation of the location of the ulnar bone as a landmark to place the electrode device. Additional detail on the device and placement is found in [1]. Data from two sessions for this dataset is provided, with the second session occurring about one hour after the first session. By performing well on leave-one-session-out tests on this dataset, we can evaluate robustness of the wearable EMG device as a control interface over timescales of around an hour after EMG control interface use. Before inclusion in the FlexWear-HD dataset, participants gave their written informed consent and agreed that this material can be used in journals and other public media. The

study protocol was approved by the Carnegie Mellon University Institutional Review Board, protocol 2021.00000121.

### A.8 Additional Results Classifying Phase-Based Information

We anticipated that phase-based featurization from each electrode's data may be able to be used to capture the spatial propagation of muscle action potentials. In order to test this, we extracted the phase from STFT and the instantaneous phase from the intrinsic mode functions of a Hilbert transform for each electrode. Results are shown in Table 11.

| Task | |
| --- | --- |
| **Pretraining before few-shot fine-tuning, LOSO-CV** | |
| Phase Spectrogram | 22.4/64.3 |
| HHT Phase | 25.1/67.1 |
| **Finetuning with first 20% of data from left-out subject** | |
| Phase Spectrogram | 27.0/69.3 |
| HHT Phase | 31.0/72.3 |

Table 11: Performance using phase-based representations for the Ninapro-DB5 dataset.

### A.9 Additional Results Using Domain Generalization Techniques

We tested typical domain generalization techniques, such as invariant risk minimization (IRM), and correlation alignment (CORAL). These techniques were tested in two previous benchmarks on domain generalization [63, 62]. The results for the Ninapro DB5 dataset is in Table 12.

| Task | |
| --- | --- |
| **Pretraining before few-shot fine-tuning, LOSO-CV** | |
| CORAL | 41.0/78.6 |
| IRM | 42.3/78.7 |
| **Finetuning with first 20% of data from left-out subject** | |
| CORAL | 68.8/93.9 |
| IRM | 68.4/93.5 |

Table 12: Performance using domain generalization-based training methods for the Ninapro-DB5 dataset.

