**Dataset Documentation and Intended Uses:**
https://huggingface.co/datasets/jehanyang/FlexWear-HD/blob/main/FlexWear-HD_DataCard.pdf

**Dataset Link:**
https://huggingface.co/datasets/jehanyang/FlexWear-HD

**Benchmark Link:**
https://emgbench.github.io

**Croissant Metadata:**
https://huggingface.co/datasets/jehanyang/FlexWear-HD/blob/main/croissant.json

**Author Statement:**
The authors hereby confirm that they bear full responsibility for any violations of rights, including but not limited to intellectual property rights, that may arise from the content of this work. Additionally, they confirm that all data used in this work is licensed appropriately and in accordance with all applicable laws and regulations.

**Hosting, Licensing, and Maintenance Plan:**
Dataset is hosted on a Hugging Face dataset repository. Licensing: CC BY 4.0. Maintenance will be limited due to the static nature of the data in the dataset. Additional information on plan is found at the data card:
https://huggingface.co/datasets/jehanyang/FlexWear-HD/blob/main/FlexWear-HD_DataCard.pdf

**Dataset DOI:**
https://doi.org/10.57967/hf/2505