# OpenReview forum: "EMGBench: Benchmarking Out-of-Distribution Generalization and Adaptation for Electromyography"
_NeurIPS.cc/2024/Datasets_and_Benchmarks_Track — NeurIPS 2024 Track Datasets and Benchmarks Poster_

### Official Review · Reviewer_qE7h · 2024-07-07
**Well presented meta-corpus of EMG gesture classification**

**Rating:** 7
**Confidence:** 4
**Correctness:** Yes
**Clarity:** Yes

**Review:**

This manuscript is a great fit for the track. It is focused on enabling benchmarking of algorithms across multiple different corpora, which I believe is the first of its kind in the EMG field. In general in the EMG field, new algorithms for analysis are presented ad hoc and this allows for some harmonization through aggregation. Moreover the manuscript focuses on generalization and adaptation, which are the two biggest challenges in the field. The manuscript provides a gentle introduction to the sEMG field for the NeurIPS audience as well.

There are some weaknesses (see below), in that the algorithms and new dataset lack a large degree of originality, and algorithms are now saturating in performance, making it perhaps unlikely that a breakthrough algorithmic approach will emerge.

**Strengths:**

- The figures have many visual storytelling elements and the manuscript overall is clear and easy to follow
- The corpus approaches the most critical problem in the EMG research field and allows for finding common ground across multiple approaches and datasets present in the field.
- Table summary of datasets in the appendix is comprehensive and useful summary for the EMG field.

**Additional Feedback:**

nit: Remove guidelines from the supplemental checklist

**Documentation:**

Yes. The authors should consider a model card or datasheet for the dataset to comply with the track's reccomendation.

**Limitations:**

The authors should consider adding a section on societal impacts.

**Opportunities For Improvement:**

- The Contribution is more of a refactoring and restatement of existing datasets, rather than entirely new datasets.
- There is not a dramatic difference in performance for algorithms across different featurizations, especially consistent changes, suggesting that there may not be large algorithmic breakthroughs to be gleaned from having a dataset and benchmark. Performance on the fine tuning task especially saturates.
- The datasets are limited in the types of gestures considered and lack challenging or in the wild conditions

**Relation To Prior Work:**

Yes

**Summary And Contributions:**

This manuscript introduces a meta-corpus of 6 surface EMG (sEMG) gesture detection datasets, and a benchmarking approach for evaluating different algorithms on generalizing across participants and ‘adapting’ (personalizing) within participants. They contribute one new high-density sEMG dataset and provide comprehensive documentation of the other 5 that are drawn from the literature, including detailed breakdowns of the number and types of gestures across datasets. They evaluate multiple different featurization methods, including raw, RMS, and time-frequency decomposed features and those drawn from 2D CNNs, comparing gesture classification accuracy and area under the curve for generalization and adaptation tasks. They compare these values across datasets, and also compare personalization performance with variable amounts of training data. They discuss differences across datasets in performance and limitations of the approach. The data and code is available.

---

> ### Author Rebuttal · Authors · 2024-08-16
>
> Thanks a lot for your constructive and encouraging feedback! We are very grateful that you recognize the importance of our proposed benchmark for the field of research for EMG. Below we address the opportunities for improvement that you mentioned.
>
> > **Opportunities for Improvement**:
> > * The Contribution is more of a refactoring and restatement of existing datasets, rather than entirely new datasets.
>
> Thank you for your suggestions for future work. We agree that the benchmark integrates existing datasets and evaluates them under unified experimental settings and preprocessing steps. We introduce a new dataset that includes data from an easy-to-wear high-density EMG device over 13 subjects; yet we acknowledge that the core contribution of emgBench is the benchmark. As part of the benchmark code release, we are also introducing a generic utils file that allows researchers to integrate their own datasets, with detailed instructions provided in the README.md in https://www.github.com/maxwellsoh/emgBenchmarking. We anticipate that researchers can compare new datasets with the current datasets in the benchmark in future work.
>
> > * There is not a dramatic difference in performance for algorithms across different featurizations, especially consistent changes, suggesting that there may not be large algorithmic breakthroughs to be gleaned from having a dataset and benchmark. Performance on the fine tuning task especially saturates.
>
> Thank you for suggesting an opportunity for future work to find what methods may improve performance overall for all datasets. As we are introducing the first benchmark involving several EMG datasets, we are optimistic that the EMG research field will develop new methods that show high generalization performance across multiple datasets. Our expanded benchmark now includes 9 diverse EMG datasets (with the addition of Ninapro DB2, Ninapro DB3, and MCS [8]). We have observed that all datasets consistently benefit from fine-tuning adaptation methods, with general trends indicating performance increases when more data from the left-out subject is included, as shown in Table 3:
>
> | Task         | Myo Dataset | UCI EMG   | Ninapro DB5 | Capgmyo   | Ninapro DB2 | Ninapro DB3 | MCS       | Hyser     | FlexWear-HD |
> |--------------|-------------|-----------|-------------|-----------|-------------|-------------|-----------|-----------|-------------|
> | **FT-20%**   | 94.9/99.4   | 91.7/99.1 | 68.6/93.6   | 89.7/98.6 | 52.4/85.8   | 44.1/80.1   | 90.7/98.9 | 83.2/96.2 | 96.9/99.8   |
> | **FT-40%**   | 98.4/99.9   | 92.2/99.4 | 76.6/96.5   | 95.9/99.7 | 55.6/88.6   | 49.4/83.0   | 95.6/99.7 | 93.0/98.8 | 98.9/99.9   |
> | **FT-60%**   | 99.0/100.0  | 95.3/99.6 | 82.1/97.8   | 96.1/99.7 | 56.0/89.1   | 50.4/84.8   | 96.8/99.7 | 92.7/99.1 | 98.7/99.8   |
> | **FT-80%**   | 99.6/100.0  | 97.0/99.7 | 77.5/96.6   | 98.1/99.9 | 56.4/89.9   | 51.8/86.8   | 97.1/99.8 | 96.4/99.4 | 99.5/100.0  |
>
> While performance is already high for certain datasets after adaptation using around 1 minute or more of data, such as for the FlexWear-HD and the Myo Dataset, we have added a new row in Table 3 that shows few-shot fine-tuning performance with 5% of the left-out subject data. Using 5% of left-out subject data corresponds to just 15 seconds of data for FlexWear-HD, and 7.5 seconds for the Myo dataset. The results show a drop in performance of 11.9% for Ninapro-DB3, 10.4% for UCI EMG, 2.7% for FlexWear-HD, and 1.8% for the Myo Dataset when compared to the previous results for fine-tuning using 20% of the left-out subject data. These findings highlight the potential for significant improvements in few-shot fine-tuning, which is particularly important for practical applications that ideally require short calibration times. We have added additional text pointing out this difference in test accuracies for 5% left-out subject data in Section 5.4.
>
> | Task         | Myo Dataset | UCI EMG   | Ninapro DB5 | Capgmyo   | Ninapro DB2 | Ninapro DB3 | MCS       | Hyser     | FlexWear-HD |
> |--------------|-------------|-----------|-------------|-----------|-------------|-------------|-----------|-----------|-------------|
> | **FT-5%**    | 93.1/99.2   | 81.3/96.3 | 48.8/83.7   | 73.8/95.9 | 37.3/76.7   | 32.2/71.2   | 80.7/96.2 | 68.4/91.7 | 94.2/99.6   |
>
>
> 8.  Mehmet Akif Ozdemir, Deniz Hande Kisa, Onan Guren, and Aydin Akan. Dataset for multi-channel surface electromyography (sEMG) signals of hand gestures. Data in brief, 41:107921, 2022.

---

> > ### Author Rebuttal · Authors · 2024-08-16
> >
> > We continue our rebuttal for Reviewer qE7h below:
> >
> > > * The datasets are limited in the types of gestures considered and lack challenging or in the wild conditions
> >
> > Thank you again for your time in reviewing the work and recommendations to improve the quality of the paper. We have now expanded the benchmark to include options for testing several additional gestures within the Ninapro datasets. Specifically, we can now test Ninapro-DB2 with data for 50 gestures, Ninapro-DB3 with 41 gestures, and Ninapro-DB5 with 53 gestures. We are including a new experiment varying the proportion of data that we fine-tune with, akin to Table 3 where we vary the proportion from 5% to 80%, and will show these results in the Appendix A.10.
> >
> > We welcome any suggestions the reviewer may have on in-the-wild datasets. The authors are not aware of any publicly available in-the-wild EMG datasets that include ground truth labels for the gestures users are actually performing. We have added new text in the Limitations and Future Work section that discusses the need for more in-the-wild datasets, and are excited to highlight this key challenge and opportunity for the field as a whole:
> >
> > “Currently, none of these datasets capture gesture data performed in real-world scenarios where users engage in everyday activities or utilize an EMG-based gesture classification system for interacting with computers or robots. Future work would benefit from the development of a dataset that includes ground truth data on gestures performed spontaneously by the user, outside of controlled environments and cue-based systems. Such a dataset would capture differences on how gestures are performed in real-world settings, improving the performance and adaptability of EMG-based gesture classification systems once properly benchmarked.”
> >
> > > **Documentation:**
> > >
> > > Yes. The authors should consider a model card or datasheet for the dataset to comply with the track's reccomendation.
> >
> > We have provided a model card for the dataset, as recommended by the track.
> >
> > https://huggingface.co/datasets/jehanyang/FlexWear-HD/blob/main/FlexWear-HD_DataCard.pdf
> >
> > > **Additional Feedback:**
> > >
> > > nit: Remove guidelines from the supplemental checklist
> >
> > Thank you for the suggestion. We agree it may appear odd to keep the guidelines in the checklist. We chose to keep them in as the instructions appear to mention that we should keep the guidelines in the supplemental checklist, possibly for easier reference by reviewers.

---

> > > ### Comment · Reviewer_qE7h · 2024-08-17
> > > **Thank you**
> > >
> > > Thanks for the rebuttal to myself and the other authors. The new, more challenging datasets improve the impact of the work. The data upload proposition is interesting. I wonder if a formal effort akin to a benchmark competition would be needed to drive engagement.
> > >
> > > I also want to highlight Reviewer Fh4V's point about treating this as an audio benchmark not a vision benchmark. The null class (no gesture) should be included in all of these classification problems.

---

> > > > ### Author Rebuttal · Authors · 2024-08-23
> > > >
> > > > We appreciate the suggestion for helping drive engagement for the benchmark. It definitely sounds interesting to try to create a benchmark competition, and it may be worth forming a competition at an upcoming conference.
> > > >
> > > > We agree that including a null (no gesture) class would also be useful, and we will investigate ways to incorporate this with the existing public datasets. As a starting point, we are including the classification of transitions windows (transitions between 2 gestures, detailed in response to Reviewer Fh4V)

---

### Official Review · Reviewer_xM2s · 2024-07-25
**An intersting benchmark studying Out-of-Distribution Generalization and Adaptation for Electromyography**

**Rating:** 6
**Confidence:** 3
**Correctness:** No.
**Clarity:** Yes

**Review:**

Please find my detailed review comments below.

**Strengths:**

(+) Considering the distribution shifts in the EMG classification task is crucial;

(+) The task and the OOD setting are presented clearly and sound;

(+) The benchmarking and analysis are comprehensive and interesting;

**Additional Feedback:**

N/A

**Documentation:**

Yes

**Limitations:**

Yes

**Opportunities For Improvement:**

1. Extending the benchmarking to the OOD generalization methods considered [1,2]. Otherwise, one will not know whether the task has already been resolved, or the remaining challenges.

2. There is only one random seed per experiment. If there is not sufficient computational resources, it's suggested to select some of representative experiments to conduct the benchmarking with more random seeds. Otherwise, the results may not be sufficiently convincing.

**References**

[1] In Search of Lost Domain Generalization, ICLR'20.

[2] WILDS: A Benchmark of in-the-Wild Distribution Shifts, ICML'21.

**Relation To Prior Work:**

Yes

**Summary And Contributions:**

This work studies the EMG classification problem, that is to predict the users' intended gestures. The authors curate a large-scale benchmark that includes 6 datasets that cover two OOD generalization scenarios: 1) intersubject classification, and 2) adaptation using train-test splits for time series. They conduct a comprehensive analysis of the distribution shifts in the benchmark, and evaluate several preprocessing methods and DNNs to demonstrate the challenges in the EMG classification problem.

---

> ### Author Rebuttal · Authors · 2024-08-16
>
> Thank you for recognizing the importance of our benchmarking work for the EMG research field. We are eager for the EMG community to benchmark the robustness of their classification models through the methods we developed for testing generalization and adaptation for realistic EMG interface applications.
>
> We address the opportunities for improvement below:
>
> > **Opportunities for Improvement:**
> >
> > * Extending the benchmarking to the OOD generalization methods considered [1,2]. Otherwise, one will not know whether the task has already been resolved, or the remaining challenges.
>
> Thank you for pointing out these methods that we could consider for generalization and distribution shifts. We see that in [1] and [2], the algorithms tested include standard supervised learning (called ERM in both [1] and [2]), invariant risk minimization (IRM), and correlation alignment (CORAL), but we do note that neither [1] or [2] found that IRM or CORAL perform better than supervised learning.
>
> These papers found supervised learning to perform as well as OOD generalization specific models. However, these results were for different datasets and we too are curious if we observe any performance increase on EMG data for IRM or CORAL.
>
> In addition to supervised learning, we are including experiments to test the IRM and CORAL methods while varying the dataset tested and the proportion of data used for adaptation, akin to the experiments in Table 3. These new results will be included in Appendix A.10 and is referenced in the main paper.
>
> > * There is only one random seed per experiment. If there is not sufficient computational resources, it's suggested to select some of representative experiments to conduct the benchmarking with more random seeds. Otherwise, the results may not be sufficiently convincing.
>
> Thank you for the suggestion to improve the quality of the benchmark. The reviewer is correct in identifying that computing the full benchmark is computationally expensive. However, we agree there is value in assessing the variance in results over multiple seeds. We are currently retraining the models used for Table 3, which we will use to report test accuracy mean and standard deviation over 3 seeds. Our expanded benchmark now includes 9 diverse EMG datasets (with the addition of Ninapro DB2, Ninapro DB3, and MCS [8]). Right now, we have results with 2 seeds for Table 3 for 8 datasets which we report in the following table:
>
> | Task         | Myo Dataset               | UCI EMG                 | Ninapro DB5            | Capgmyo                 | Ninapro DB2            | Ninapro DB3            | Hyser                   | FlexWear-HD            |
> |--------------|---------------------------|-------------------------|------------------------|-------------------------|------------------------|------------------------|-------------------------|------------------------|
> | **FT-20%**   | 95.15 (0.35) / 99.45 (0.07)| 91.5 (0.28) / 99.1 (0.00)| 69.15 (0.78) / 93.9 (0.42)| 89.7 (0.00) / 98.75 (0.21)| 52.15 (0.35) / 85.95 (0.21)| 44.3 (0.28) / 80.3 (0.28)| 84.4 (1.70) / 96.4 (0.28)| 96.75 (0.21) / 99.75 (0.07)|
> | **FT-40%**   | 98.35 (0.07) / 99.9 (0.00) | 92.45 (0.35) / 99.35 (0.07)| 76.3 (0.42) / 96.5 (0.00)| 96.15 (0.35) / 99.7 (0.00)| 55.9 (0.42) / 88.75 (0.21)| 49.45 (0.07) / 83.15 (0.21)| 91.4 (2.26) / 98.4 (0.57)| 98.85 (0.07) / 99.9 (0.00)|
> | **FT-60%**   | 98.95 (0.07) / 100.0 (0.00)| 95.0 (0.42) / 99.6 (0.00)| 82.0 (0.14) / 97.75 (0.07)| 96.45 (0.49) / 99.8 (0.14)| 56.6 (0.85) / 89.2 (0.14)| 50.9 (0.71) / 84.9 (0.14)| 92.35 (0.49) / 98.65 (0.64)| 98.95 (0.35) / 99.8 (0.00)|
> | **FT-80%**   | 99.55 (0.07) / 100.0 (0.00)| 97.7 (0.99) / 99.8 (0.14)| 78.05 (0.78) / 96.9 (0.42)| 97.55 (0.78) / 99.85 (0.07)| 55.8 (0.85) / 89.95 (0.07)| 52.85 (1.48) / 87.25 (0.64)| 96.75 (0.49) / 99.05 (0.49)| 99.45 (0.07) / 100.0 (0.00)|
>
> The standard deviation between two seeds is small for most of these results. The only standard deviations above 1% comes from the Hyser dataset after fine-tuning with 20% and 40% of the leftout subject’s data, with a standard deviation of 1.7% and 2.3%, respectively.
>
> 8.   Mehmet Akif Ozdemir, Deniz Hande Kisa, Onan Guren, and Aydin Akan. Dataset for multi-channel surface electromyography (sEMG) signals of hand gestures. Data in brief, 41:107921, 2022.
>
> > **Correctness:**
> >
> > No.
>
> Thank you for your time that you have spent reviewing our work in detail. Let us know if the above additions to the results in the paper adequately addresses where you believe there are correctness issues. We are very eager to improve the quality of our work with your feedback.

---

> > ### Comment · Reviewer_xM2s · 2024-08-21
> >
> > Thanks the authors for the rebuttal and additional experiments. I would like to maintain my overall positive rating of this paper if the promised experiments will be added to the paper (to the main paper would be much appreciated).

---

> > > ### Author Rebuttal · Authors · 2024-08-23
> > >
> > > We thank the reviewer for the additional comment. We will add all additional experiments to the paper. We will be sure to add additional results to the main paper for all results that we can fit in the page limit.

---

### Official Review · Reviewer_Fh4V · 2024-07-25
**Gesture EMG classification across multiple public datasets using deep vision models**

**Rating:** 6
**Confidence:** 5
**Correctness:** very likely to be correct.
**Clarity:** very clear

**Review:**

The well is ambitious covering many dataset and comparing many vision architectures. I however have a few concerns:

- problem framing: the paper insists on making the benchmark realistic for real world applications although by design it is not really an answer to a real scenario. Indeed the benchmark uses windows of data around gestures so it starts by assuming that one knows when gestures occur. To be realistic you need to able to predict from a stream of data when a gesture occurs and what it is. Treating the problem with a computer vision angle and not a problem on a data stream eg for audio event detection seems problematic.

- it seems that phase information between electrodes is important to be sensitive to the muscle action potential although the features proposed do not seem able to capture this.

- please report the number of parameters for each tested architectures

- the model training is not blind to the new domain as it is actually used for early stopping. This is to me a be a big issue as it's another evidence that the proposed benchmark is not realistic and cannot be used as is as reference benchmark for the field.

**Strengths:**

- good writing
- very solid literature review although this recent paper could be cited https://www.biorxiv.org/content/10.1101/2024.02.23.581779v1

**Additional Feedback:**

fix the references so that all acronyms are with capita letters. You should put them between {} in the bibtex.

**Documentation:**

very good.

**Ethics:**

no concern

**Limitations:**

- not a realistic benchmark as is

**Opportunities For Improvement:**

- fix the model training overfitting that uses the left out subject for early stopping

**Relation To Prior Work:**

very good.

**Summary And Contributions:**

Website: https://emgbench.github.io
Code: https://github.com/maxwellsoh/emgBenchmarking

This paper addresses the problem of gesture classification from surface EMG data. This problem is cast as a domain generalization problem (where new domain = new subject) or a few shot learning problem where about 1 min of data for a new subject is used for model fine-tuning / calibration. The paper aims to become a standard benchmark for EMG data.

---

> ### Author Rebuttal · Authors · 2024-08-16
>
> We want to extend our heartfelt gratitude for taking the time to review our paper. Thank you for your valuable comments and suggestions on improving the quality of the paper.
>
> > **Review:**
> >
> > The well is ambitious covering many dataset and comparing many vision architectures. I however have a few concerns:
> >
> > * problem framing: the paper insists on making the benchmark realistic for real world applications although by design it is not really an answer to a real scenario. Indeed the benchmark uses windows of data around gestures so it starts by assuming that one knows when gestures occur. To be realistic you need to able to predict from a stream of data when a gesture occurs and what it is. Treating the problem with a computer vision angle and not a problem on a data stream eg for audio event detection seems problematic.
>
> Based on this comment, we will add new results in the appendix using two methods that could account for these transitions.
>
> 1. Training of an additional binary classifier that decides whether a window is a transition period or not. Given a small step size between windows, this binary classifier could be used to determine when a transition happens with the resolution of the step size, allowing the researcher to decide how they should use the gesture classifier during this transition phase.
> 2. Training new gesture classifiers that are trained with windows during the transition periods from one gesture to another. Although windows of data during these transition periods may have multiple labels, we would label these transition windows using the label from the last time step of the sample window. This approach assumes that the system should begin adapting to the next gesture during the transition, enabling a smoother and more responsive user experience.
>
> Of the 6 current datasets, only 2 provide transition data: Ninapro-DB5 and UCI EMG. To further evaluate the transition states between gestures, we are adding 3 new datasets: Ninapro-DB2, Ninapro-DB3, and MCS [2]. We are currently gathering results to add these datasets to all the result tables, including the new results on transitions in the appendix. We also add this text into the Limitations and Future Work section:
>
> “Datasets such as CapgMyo, Hyser, Myo Dataset, and FlexWear-HD do not present transition data between gestures, likely due to the added complexity of labeling short dynamic transition data for a gesture-based classification interface. However, we note that in previous robot control work with the FlexWear-HD [1], while these transition windows were also not taken into account during training, users were able to control a robot using streams of EMG data classified in real-time to perform complex assistive tasks through teleoperation. However, for the datasets that do include transition data, such as the Ninapro datasets, MCS, and UCI EMG, we include additional results in Appendix Section A.8 where the transition windows are classified using two different methods.”
>
> 1. Jehan Yang, Kent Shibata, Douglas Weber, and Zackory Erickson. High-density electromyography for effective gesture-based control of physically assistive mobile manipulators. arXiv: preprint arXiv:2312.07745, 2023
> 2.   Mehmet Akif Ozdemir, Deniz Hande Kisa, Onan Guren, and Aydin Akan. Dataset for multi-channel surface electromyography (sEMG) signals of hand gestures. Data in brief, 41:107921, 2022.

---

> > ### Author Rebuttal · Authors · 2024-08-16
> >
> > We continue our response to the rest of the comments from Reviewer Fh4V below:
> >
> > > * it seems that phase information between electrodes is important to be sensitive to the muscle action potential although the features proposed do not seem able to capture this.
> >
> > Thank you for the suggestion. We agree that phase information may be used to capture the propagation of muscle action potentials. In order to test whether phase information may be useful in classification for our benchmark, we added two additional preprocessing features where 1) phase is extracted from the short-term-fourier transform, and 2) instantaneous phase is extracted from the intrinsic mode functions for a Hilbert transform [29], the results of which we will include in the appendix. We will include both generalization and adaptation results for these preprocessing methods, in the same style as Table 1, in Appendix A.9.
> >
> > > * please report the number of parameters for each tested architectures
> >
> > Thank you for pointing out this suggestion. We have listed the number of parameters for each tested architecture in the Results Section 5.3 where different model architectures are introduced.
> >
> > “The number of parameters used for each model is 2 million for EfficientViT, 4 million for EfficientNet, 6 million for ViT, and 11 million for Resnet18.”
> >
> > > * the model training is not blind to the new domain as it is actually used for early stopping. This is to me a be a big issue as it's another evidence that the proposed benchmark is not realistic and cannot be used as is as reference benchmark for the field.
> >
> > Thank you for your suggestion to help improve the quality of the paper. However, no models in our work use early stopping or any left out subject data in any form during training. We apologize for any confusion here. We have added additional text in Section 4 and Section 5 of the paper to further reinforce this.
> >
> > **Section 4:**
> >
> > “We use constant numbers of epochs for all models, and the tables presented are based on test accuracies from these models that are always trained for a constant number of epochs.”
> >
> > **Section 5:**
> >
> > “For these experiments, pretraining is conducted over 100 epochs and fine-tuning is conducted over 750 epochs.”
> >
> > > **Strengths:**
> > >
> > > * good writing
> > > * very solid literature review although this recent paper could be cited https://www.biorxiv.org/content/10.1101/2024.02.23.581779v1
> >
> > Thank you for the positive feedback on the clarity of the writing of the paper. We are happy that the paper is clear. We are eager for the EMG and machine learning communities to be able to adopt this benchmark to help improve the robustness of classification models for realistic EMG interface applications.
> >
> > We have referenced this recent paper in the related work for EMG decoding, noting that it is a large-scale generalization and adaptation paper for EMG. Currently, this paper uses a device that is not commercially available and does not publish a public dataset, but the scale of the work is very impressive. We note that the trend this work found with personalizing an EMG model by fine-tuning resulted in a greatly improved performance in Section 5.4, similar to in our study:
> >
> > “A large-scale study by Sussillo et al. found a similar trend, where personalizing a model by fine-tuning improved performance on EMG tasks by 30%.”
> >
> > > **Opportunities for Improvement**
> > >
> > > * fix the model training overfitting that uses the left out subject for early stopping
> >
> > Thank you for this suggestion to improve the paper. As mentioned above, we actually do not use any form of early stopping, and instead always use a constant number of time steps for training. We have added additional text in Section 4 and Section 5 of the paper to clarify any possible confusion on early stopping.
> >
> > **Section 4:**
> >
> > “We use constant numbers of epochs for all models, and the tables presented are based on test accuracies from these models that are always trained for a constant number of epochs.”
> >
> > **Section 5:**
> >
> > “For these experiments, pretraining is conducted over 100 epochs and fine-tuning is conducted over 750 epochs.”
> >
> > > **Limitations:**
> > >
> > > not a realistic benchmark as is
> >
> > We thank you for your highly constructive feedback and taking the time to review our paper. As much as possible, we would like for our work to be used as a realistic benchmark for realistic EMG interface applications. Please let us know if we addressed your concerns adequately or how we may further improve our paper by the end of the discussion period.
> >
> > > **Additional Feedback:**
> > >
> > > fix the references so that all acronyms are with capita letters. You should put them between {} in the bibtex.
> >
> > Thank you for catching this. We have fixed the references so that all acronyms are with capital letters.

---

> > > ### Comment · Reviewer_Fh4V · 2024-08-20
> > >
> > > I appreciate the effort conducted during the rebuttal and I am bumping more score to 6 based on the planned changes.

---

### Author Rebuttal · Authors · 2024-08-16

We thank all the reviewers for their highly constructive feedback. We are pleased that all the reviewers found the paper very clear and recognized our thorough literature review. We also appreciate that our efforts to relate our work to previous research were noted, as we believe this to be of utmost importance for a new benchmarking paper on EMG.

Based on the reviewers' feedback, we have made several significant additions to the paper:

* New Datasets: We have added three new datasets, bringing in a total of 91 additional subjects. These datasets include dozens of gestures and transition data [1, 2].
* Benchmark Enhancements: We introduced new options in the benchmark to train with transition data between gestures for the 5 out of 9 datasets that provide it (Appendix A.8).
* Feature Extraction: We included phase-based feature extraction methods (Appendix A.9).
* Method Evaluations: We evaluated invariant risk minimization (IRM) and correlation alignment (CORAL) methods (Appendix A.10).
* Robustness Assessment: We used multiple seeds to evaluate the mean and standard deviation for average test accuracy and average test area under the receiver operating curve (AUC) for Table 3.
* Fine-tuning: We applied fine-tuning with 5% of the data from the left-out subject, representing just 15 seconds of data when looking at the FlexWear-HD dataset.

We note that due to computational constraints, the final results for all these additional analyses are still being computed and will be included in the final version of the paper.

We look forward to engaging in further discussions with the reviewers to enhance the quality of our paper, ensuring it serves as a realistic and useful benchmark for EMG classification in the future. Thank you for your invaluable time and feedback.

References:

1.   Mehmet Akif Ozdemir, Deniz Hande Kisa, Onan Guren, and Aydin Akan. Dataset for multi-channel surface electromyography (sEMG) signals of hand gestures. Data in brief, 41:107921, 2022.

2. Manfredo Atzori, Arjan Gijsberts, Simone Heynen, Anne-Gabrielle Mittaz Hager, Olivier Deriaz, Patrick Van Der Smagt, Claudio Castellini, Barbara Caputo, and Henning Müller. Building the Ninapro database: A resource for the biorobotics community. In 2012 4th IEEE RAS & EMBS International Conference on Biomedical Robotics and Biomechatronics (BioRob), pages 1258–1265. IEEE, 2012.538

---

### Decision · Program_Chairs · 2024-09-26

**Decision:**

Accept (Poster)

**Comment:**

This paper presents the inaugural benchmark for evaluating the out-of-distribution performance of electromyography (EMG) classification algorithms via machine learning. It encompasses six datasets, comprehensive evaluation process, and novel options. This benchmark addresses the critical shortage of open-source resources within the EMG research domain and provides an essential tool for investigating the practical dimensions of out-of-distribution performance. The reviewers are unanimous in their assessment that this contribution holds significant value for the community. The authors have addressed all comments and concerns raised. The paper is recommended for acceptance.